# A Multi-Scale Fusion Strategy for Side Scan Sonar Image Correction to Improve Low Contrast and Noise Interference

**Ping Zhou** [1] , **Jifa Chen** [2,*] , **Pu Tang** [1] , **Jianjun Gan** [1,3] and **Hongmei Zhang** [1]

1 School of Hydraulic Engineering, Nanchang Institute of Technology, Nanchang 330099, China; pingzhou@cug.edu.cn (P.Z.); 2003992816@nit.edu.cn (P.T.); ganjianjun@nit.edu.cn (J.G.); 2006992934@nit.edu.cn (H.Z.)
2 Key Laboratory of Poyang Lake Wetland and Watershed Research, Ministry of Education, Jiangxi Normal University, Nanchang 330022, China
3 Faculty of Engineering, University of Alberta, Edmonton, AB T6G 1H9, Canada
* Correspondence: chenjifa@cug.edu.cn; Tel.: +86-134-2988-0845

**Abstract:** Side scan sonar images have great application prospects in underwater surveys, target detection, and engineering activities. However, the acquired sonar images exhibit low illumination, scattered noise, distorted outlines, and unclear edge textures due to the complicated undersea environment and intrinsic device flaws. Hence, this paper proposes a multi-scale fusion strategy for side scan sonar (SSS) image correction to improve the low contrast and noise interference. Initially, an SSS image was decomposed into low and high frequency sub-bands via the non-subsampled shearlet transform (NSST). Then, modified multi-scale retinex (MMSR) was employed to enhance the contrast of the low frequency sub-band. Next, sparse dictionary learning (SDL) was utilized to eliminate high frequency noise. Finally, the process of NSST reconstruction was completed by fusing the emerging low and high frequency sub-band images to generate a new sonar image. The experimental results demonstrate that the target features, underwater terrain, and edge contours could be clearly displayed in the image corrected by the multi-scale fusion strategy when compared to eight correction techniques: BPDHE, MSRCR, NPE, ALTM, LIME, FE, WT, and TVRLRA. Effective control was achieved over the speckle noise of the sonar image. Furthermore, the *AG*, *STD*, and *E* values illustrated the delicacy and contrast of the corrected images processed by the proposed strategy. The *PSNR* value revealed that the proposed strategy outperformed the advanced TVRLRA technology in terms of filtering performance by at least 8.8%. It can provide sonar imagery that is appropriate for various circumstances.

**Keywords:** sonar image correction; side scan sonar; non-subsampled shearlet transform; retinex theory; dictionary learning



## 1. Introduction

The increasing frequency of marine surveys and engineering activities is closely related to the large-scale and refined exploration of underwater information. Thanks to the advantages of a high imaging resolution and comprehensive coverage, side scan sonar (SSS) has become widely employed in target identification, seafloor seismic analysis, and sediment monitoring [1,2]. In addition, SSS images are used for underwater pipeline detection, cable laying, and safety assurance in underwater engineering [3]. However, the recorded SSS images exhibit low illumination and speckle noise due to the attenuation of echo signals, complicated environmental factors, and changes in the detection depth [4,5]. These abnormalities have the potential to quickly cause errors in judgment, which is highly detrimental to later scientific and technical applications. Thus, there is a pressing need to determine how to efficiently reduce the noise and improve the visual effect.

Many studies have applied radiation distortion techniques to address the issue of low light in sonar images. Capus examined how the beam direction affected side scan sonar

images' non-uniformity [6]. Zhao et al. considered the effect of sediment changes on signal radiation distortion and established a linear relationship between distortion and the sonar height, eliminating the distortion problem along the trajectory direction [7]. Combining the priors of the SSS imaging process, low-rank constraints have been introduced for the sonar illumination component. Li et al. proposed a total variation regularized low-rank approximation (TVRLRA) model to correct the low illumination problem caused by radiation distortion [8]. Al Rawi et al. used mixed exponential regression analysis to compensate for low-value pixels [9]. The energy distribution function of the beam angle also serves as a correction factor to a certain extent [10]. This type of approach necessitates the consideration of a number of variables, including the surface sediment types, seabed topography, and acoustic environment. Higher standards are necessary under specific circumstances; otherwise, the image correction effect will be subpar.

The spatial domain method aims to directly act on the sonar-scattered signal and compare it to the gray value. Such methods include Brightness Preserving Dynamic Histogram Equalization (BPDHE) [11], unsharpened masks [12], partial differential equation enhancement (PDEE) [13], etc. The BPDHE model improves the global contrast of images, but its correction and improvement capabilities are limited. The question of how to select the filter size in the unsharpened mask algorithm has still not been solved. The nonlinear requirements of partial differential equations are complex, and the actual images are unable to meet the corresponding conditions.

Furthermore, certain deep network and machine learning models have been utilized to enhance the sonar image quality. Xu et al. employed a filtering level set model to weaken the target background shadow [14]. Nevertheless, the background data surrounding the middle are not fully taken into account. Kim et al. processed several continuous sonar images using a denoising autoencoder (DAE) technique in order to address the noise correction of the original sonar data [15]. Moreover, Liu et al. presented a two-stage convolutional autoencoder (TCAE) method that allows a low-frequency sonar image to reach a resolution comparable to that of high-frequency sonar images [16]. Thomas et al. investigated the generation of a set of super-resolution sonar images through the application of generative adversarial networks (GAN) and transfer learning, successfully integrating and correcting multi-source sonar images [17]. In order to address the resilience of sonar images under turbidity and fluctuating lighting conditions, Liu and Wang employed the CycleGAN model to mimic the production of forward-looking side scan sonar image sets [18]. While some machine learning techniques can produce superior corrective effects, we cannot disregard the pragmatic concerns of time consumption and data capacity.

The side scan sonar image correction strategy based on the transform domain decomposes the image into low-frequency and high-frequency sub-bands for feature analysis. This mainly includes the Laplace pyramid transform (LP) [11], wavelet transform (WT) [19], curvelet transform (curvelet) [20], contourlet [21], and non-subsampled contourlet transform (NSCT) [22]. Among them, side scan sonar images' high-dimensional variation information cannot be reflected by the LP and WT approaches. The manual parameter settings required for curvelets and contourlets may result in the deterioration of the sub-band information in various layer locations. It is necessary to improve the NSCT's multi-scale decomposition processing efficiency. Research has found that image blurring and poor contrast in various types of images can be significantly improved by the shearlet transform and the enhanced non-subsampled shearlet transform (NSST) [23,24]. In addition, the NSST achieves better contrast enhancement and clear target contours. As a result, the NSST framework can be applied for the correction and enhancement of side scan sonar images.

When dealing with dark sonar images, retinex theory can be utilized to improve the contrast information of target contours and geographic textures. Common upgraded versions of retinex include naturalness preserving enhanced (NPE) [25], adaptive local tone mapping (ALTM) [26], the low light illumination map estimation model (LIME) [27], and multi-scale retinex with color restoration (MSRCR) [28]. Ye et al. optimized the parameters of retinex to achieve an information stretching display in the back shadow

area [29]. More instances are still required to confirm the choice of regulatory elements and their generalizability. Muthuraman added edge preservation techniques to further improve the edge clarity of the targets [5]. However, further research is needed to explore the deformation and enable the improvement of retinex.

Furthermore, the assessment of the noise distribution and the features of speckle noise can be fully taken into account when processing high-frequency sub-band images for noise. A few filtering techniques are used in the high-frequency sub-bands to suppress noise. Tang et al. utilized the Kalman filter algorithm to suppress Gaussian noise in side scan sonar images and preserve the edge details to the maximum extent possible [30]. Innocentini et al. employed morphological edge detection to achieve the denoising of sonar images under a low signal-to-noise ratio [31]. Chen et al. conducted the best estimation from the perspective of the maximum a posteriori probability method and developed a fully variational adaptive underwater sonar image denoising approach [32]. Referring to the characteristics of captured sonar images, experiments have found that side scan sonar images have inherent sparse structure characteristics [33]. Therefore, sparse dictionary learning (SDL) can effectively distinguish useful information from noise.

Overall, the presented correction techniques have enabled improvements in global contrast enhancement, noise reduction, and edge contours, as shown in Figure 1. Numerous technologies are employed to address a single aspect of a phenomenon, whereas only a small number of aspects are fully taken into account. Furthermore, real-time sonar image rectification needs to consider difficulties like the data volume and time consumption.

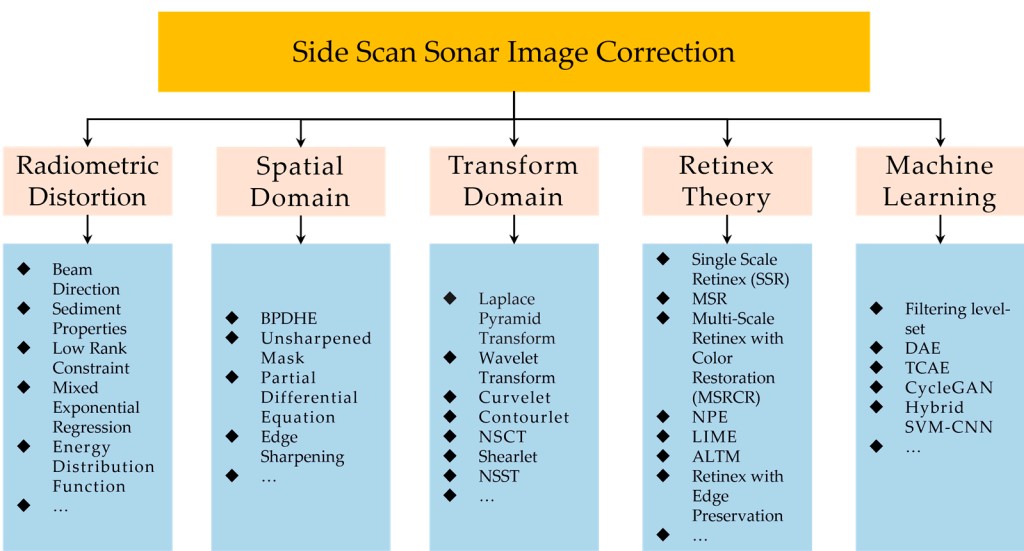

**Figure 1.** Classification review of side scan sonar image correction.

Therefore, this paper proposes a strategy of integrating multiple technologies to improve the low illumination and high noise issues in side scan sonar images. This approach improves the corresponding modules to adjust to the characteristics of side scan sonar images by combining the concept of the transformation domain with an understanding of retinex theory. The primary contributions of our suggested fusion strategy can be described as follows:

(1) The non-subsampled shearlet transform is able to achieve the multiple decomposition of sonar image signals, reducing image quality issues caused by motion blur, radiation distortion, noise, etc.

(2) The modified multi-scale retinex algorithm is applied to enhance the inherent detection energy from low-frequency sub-band images with masked information.

(3) Sparse dictionary learning helps to reduce the effects of high-frequency distortion, noise interference, and signal redundancy stacking by representing high-frequency images as a linear combination of as few non-zero coefficients as feasible.

The detailed structure of this article is as follows. Section 2 introduces the proposed multi-scale strategy and execution process. The experimental results obtained from different perspectives are shown in Section 3. Some open-source datasets and measured data were used to further validate the performance of the proposed technology, as shown in Section 4. Finally, the conclusions can be found in Section 5.

## 2. Materials and Methods

### 2.1. Overall Framework

In order to improve the sonar image performance at low light levels and reduce noise interference, this paper considers the features of sonar images from the perspectives of multi-scale fusion and retinex theory. The framework structure of the recommended technique is illustrated in Figure 2. The main components of our strategy include (1) the feature decomposition of the NSST transform; (2) the low-frequency enhancement of modified multi-scale retinex (MMSR); (3) high-frequency signal filtering using sparse dictionary learning (SDL). The performance of the correction strategy is evaluated in terms of subjective visual effects and objective evaluation indicators.

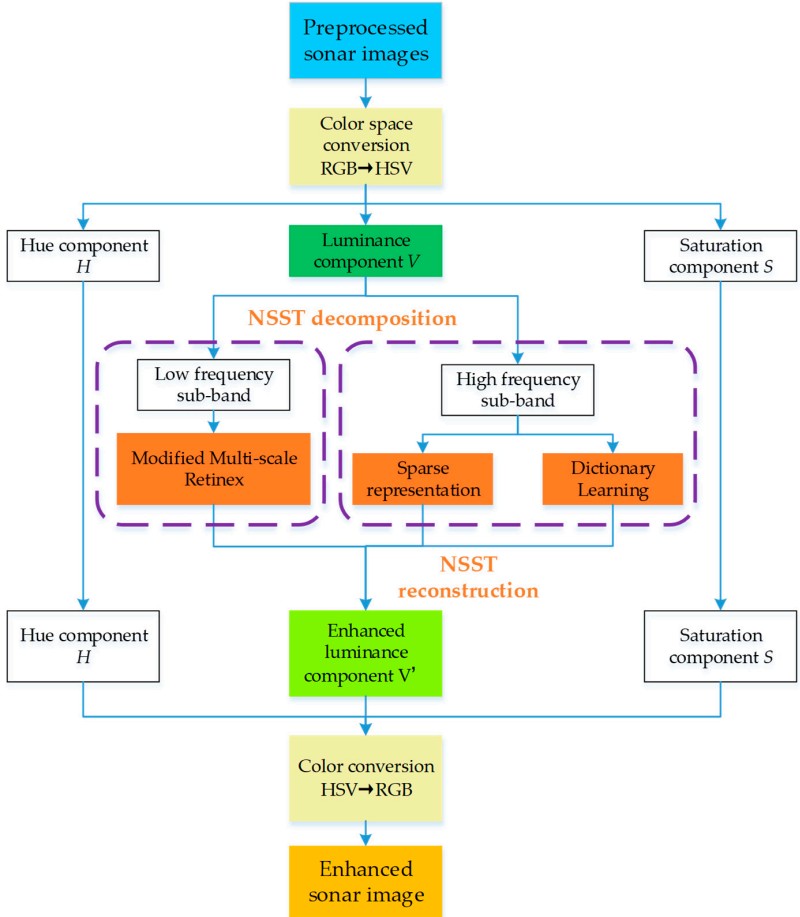

**Figure 2.** Flow chart of proposed strategy for low illumination enhancement and noise suppression.

### 2.2. Non-Subsampled Shearlet Transform

Shear waves serve as the foundation for the non-subsampling shearlet transform (NSST). Translation invariance during image decomposition is a property of the NSST [20]. Because the NSST provides an ideal sparse representation and is direction-independent, it is able to adapt to attenuation changes in the edge information and capture the multi-dimensional geometric aspects of the objects in sonar images. The 3-layer NSST decomposition process is depicted in Figure 3.

Initially, the luminance image *V* after HSV spatial transformation is decomposed into low- and high-frequency sub-bands using a non-subsampled Laplacian pyramid filter bank. Then, an improved shear wave filter is employed to achieve the directional localization of the high-frequency sub-band image. Subsequently, the secondary low-frequency and high-frequency images are gradually extracted from the low-frequency sub-band image. Ultimately, *K*-level decomposition is attained through iterative processing, yielding a total of *K* high-frequency sub-band images and 1 low-frequency image. Additionally, the original image dimensions are maintained for both the high- and low-frequency sub-band images. The inverse transformation process of the NSST is the opposite of the above.

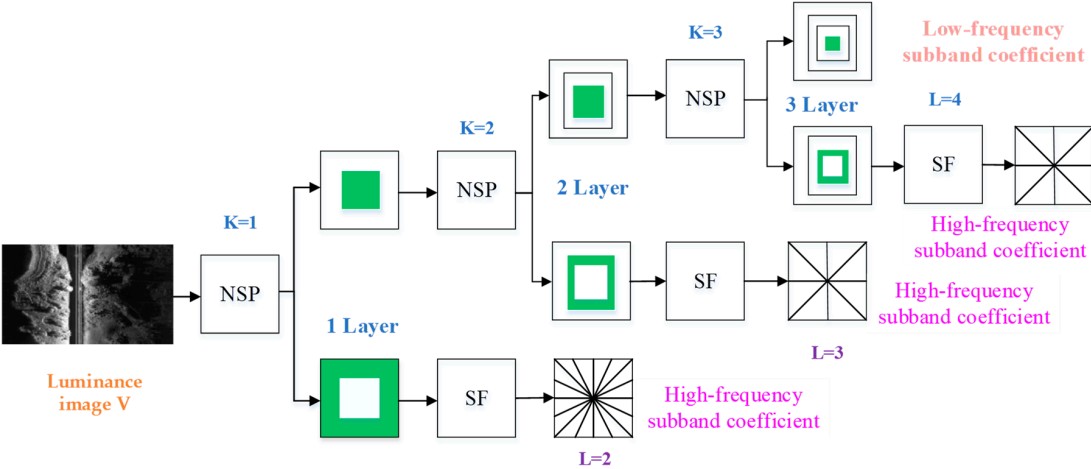

Notes: (1) NSP: Non-subsampled laplacian pyramid filter bank; (2) SF: Shear filter; (3) K represents the number of decomposition layers, which are 1,2,3.; (4) L represents the decomposition direction, which are 1, 2, and 3, respectively.

**Figure 3.** Multi-scale decomposition process of NSST with side scan sonar images.

### 2.3. Modified Multi-Scale Retinex

The low-frequency sub-band is noise-free but still contains a large amount of energy information from the original image following the breakdown of the luminance image transformation. The dark contrast and strong interference from the target shadow area are the two main features of low-frequency images. Therefore, a modified multi-scale retinex model (MMSR) is used to improve the hierarchy and dark contrast of the low-frequency sub-band image in order to ensure the balance and continuous change of the global grayscale, as shown in Figure 4.

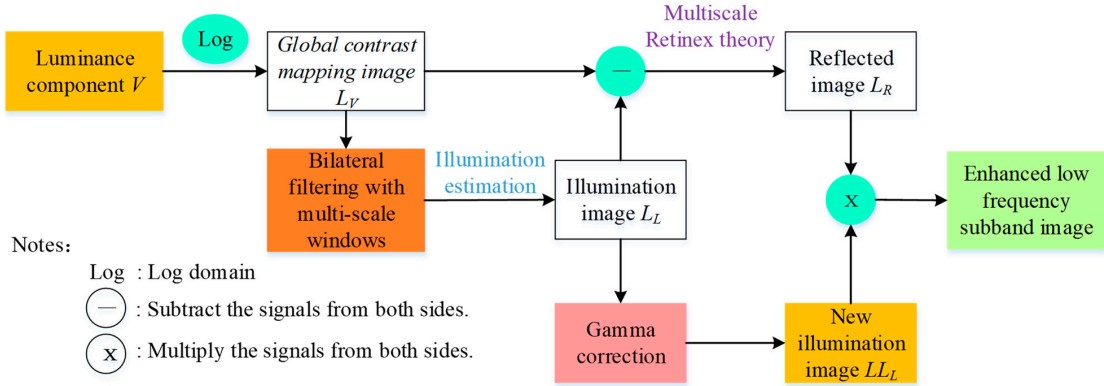

**Figure 4.** Enhancement process of low-frequency sub-band image.

Retinex is an algorithm for the improvement of the image contrast. The retinex concept states that the reflection signal of the target is combined with the illumination of the environment to form an echo signal image. Thus, our challenge is to eliminate the

illuminance effect while maintaining the intrinsic qualities of the objects in sonar images. The initial sonar brightness signal $L_V$ can be broken down into

$$L_V(x,y) = L_L(x,y) \times L_R(x,y) \tag{1}$$

where the brightness signal is denoted by $L_V$, the illuminance echo image by $L_L$, and the reflected echo signal by $L_R$.

The computation in the logarithmic domain exhibits a high degree of consistency with the nonlinear changing features of the contrast between objects visible to the human eye. As a result, we take the logarithm on both sides. The target reflected echo signal is transformed into

$$L_R = \exp(\ln(L_V + \varepsilon) - \ln(L_L + \varepsilon)) \tag{2}$$

We apply multi-scale variables to balance the processing of the reflected signals in order to address the issues of excessive contrast augmentation and color distortion.

$$L_R = \frac{1}{wk}\sum_{i=1}^{wk} \exp(\ln(L_V + \varepsilon) - \ln(L_L + \varepsilon)) \tag{3}$$

where $\omega k$ is the number of scales. Numerous tests have demonstrated that the local aspects of the image may be guaranteed by employing three large, medium, and small scales.

Moreover, retinex theory's implementation requires a reliable means of guaranteeing the original brightness feature's signal quality. For $L_V$ signals, we modify the mapping of the global grayscale values in the log domain in order to synchronize the contrast and detail consistency in the overall image. The global grayscale contrast transformation is

$$L_V(x,y) = \frac{\ln(V(x,y)/V_m + 1)}{\ln(V_{\max}/V_m + 1)} \tag{4}$$

where $V(x, y)$ is the input value of the brightness component. $V_{\max}$ represents the maximum value of the brightness component. $\varepsilon = 0.001$ in this article, to guarantee factors larger than 0 in the logarithmic domain. $V_m$ is the brightness value in the average logarithmic domain, expressed as

$$V_m = \exp\left(\frac{1}{N}\sum_{x,y}\ln(\varepsilon + V(x,y))\right) \tag{5}$$

Retinex theory uses the $L_V$ value that is optimized by the global equilibrium as the echo input signal.

Effective estimation is essential in acquiring the target's genuine reflection signal for the sonar's illuminance component ($L_L$). The multi window bilateral filter is used in this article to estimate the illumination component. The particular manifestations of this filter, which considers the spatial position of the calculated points and pixel grayscale values, are as follows:

$$L_L(p) = \frac{1}{w(p)}\sum_{q\in\Omega} L_V(q)f_1(p - q)f_2(L_V(p) - L_V(q)) \tag{6}$$

$$w(p) = \sum_{q\in\Omega} f_1(p - q)f_2(L_V(p) - L_V(q)) \tag{7}$$

where $p$ is the illuminance value of the target point, $\Omega$ is its neighborhood range, and $q$ is a point in its neighborhood. $f_1$ and $f_2$ are Gaussian kernel functions, defined as

$$f_1 = e^{-\frac{1}{2}\left(\frac{d(p,q)}{\sigma_d}\right)^2}, \; f_2 = e^{-\frac{1}{2}\left(\frac{\delta(L_V(p),L_V(q))}{\sigma_r}\right)^2} \tag{8}$$

where $d(p, q)$ is the Euclidean distance between two pixels. $\sigma_d$ is the standard deviation of the distance within the neighborhood, which characterizes the edge details and clarity of the image. The smaller the value of $\sigma_d$, the greater the required edge details and blur of the image. $\delta(L_V(p), L_V(q))$ represents the difference in the brightness values between two points.

$\sigma_r$ is the standard deviation of the brightness within the neighborhood, representing the required level of noise in the image. Owing to the experiment's strong scattering sonar image properties, $\sigma_d = 3$, $\sigma_r = 0.1$.

We select multiple kernel windows to estimate the region of the neighborhood sub-window image that preserves the information, and we take their average to represent the illuminance component of the image.

$$L_L = \frac{1}{w_h} \sum_{i=1}^{w_h} L_L(\Omega = i) \tag{9}$$

where $\omega_h$ is the number of kernel windows. In the experiment, $\omega_h = 3$. The corresponding neighborhood $\Omega$ ranges are 5, 8, and 15, respectively.

Additionally, nonlinear pre-compensation stretching is carried out on the image of the illuminance component using gamma correction after it has been obtained. The image $LL_L$ of the new illumination component is given as

$$LL_L = L_L^{(1/\gamma)} \tag{10}$$

where the gamma correction factor is denoted by $\gamma$. When $\gamma > 1$, the gray value of the low-gray area is improved as a whole, which promotes an enhancement in contrast. This characteristic is highly suitable when dealing with the dark characteristics of low-frequency sub-bands. In the experiment, $\gamma = 1.8$.

Finally, the enhanced low-frequency sub-band image $L_z$ is calculated, and $L_z$ is represented as

$$L_z = LL_L \times L_R \tag{11}$$

In order to accomplish basic grayscale balance, $L_Z$ uses a histogram truncation method to cut off grayscale values with a given likelihood of occurrence at both ends, compressing the remaining pixel values to the range of [0, 1].

### 2.4. Sparse Dictionary Learning

The high-frequency sub-bands decomposed by the NSST contain most of the target edges and texture details of the source image. In addition, residual noise from the pre-processing of the original image is present in the high-frequency sub-bands. An uneven distribution on the seabed may have an impact on this speckle noise, which are mostly created by sediment echo signals connected to the seabed's sediment background. Therefore, one of the main challenges in high-frequency sub-band processing is to properly separate noise from important or salient signals. In sonar image rectification, denoising is an essential step. Otherwise, an increase in contrast will also accentuate the noise effect. As seen in Figure 5, this article employs sparse dictionary learning (SDL) to reduce the noise interference of high-frequency sub-band images.

Firstly, we adopt an additive Gaussian noise model in the logarithmic domain. The operation of multiplying the normalized high-frequency sub-band coefficients by 255 is implemented to fully utilize the easier addition process. Secondly, *m* samples with a size of *N* windows are selected at random. A linear sparse approach is adopted to estimate new samples. Next, in order to solve the challenge of obtaining *D* and *X* variables in unconstrained optimization, we apply the overcomplete technique of the discrete cosine transform (DCT) [34] to construct the initial dictionary $D_0$. The orthogonal matching pursuit (OMP) algorithm [35] is used by the associated starting sparse matrix $X_0$ to carry out sparse encoding on the image blocks. The two optimization variables, $d_k$ and $X_T^k$, are solved for the use of singular value decomposition (SVD) algorithms. Finally, all dictionary atoms and sparse matrix vectors are iteratively updated until the corresponding set number of times or the target residual error extremum is reached. The new high-frequency sub-band image is created by performing the logarithmic inverse transformation on the new dictionary and sparse matrix. Detailed theoretical information can be referred to in the literature [36].

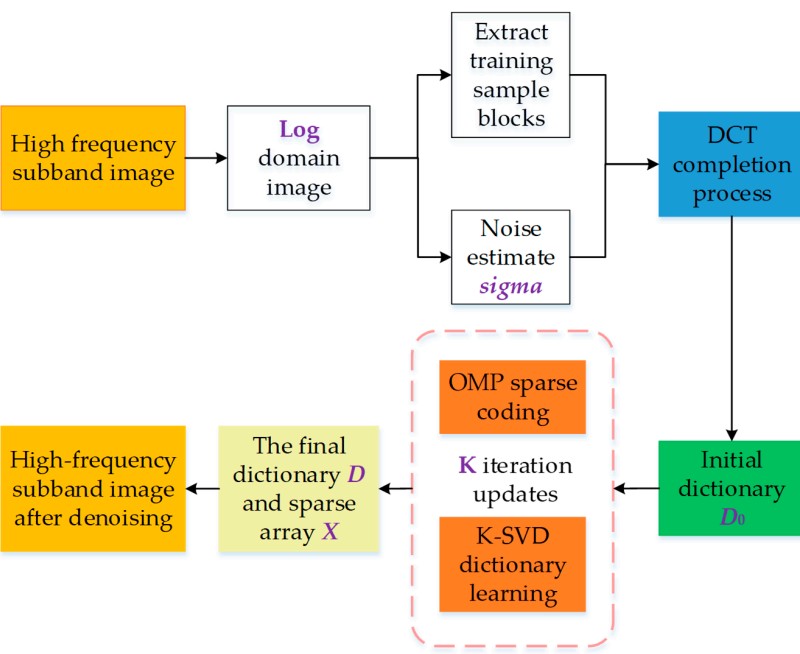

**Figure 5.** Denoising process of high-frequency sub-band image with sparse dictionary learning.

*2.5. Implementation of Proposed Strategy*

The detailed process of multi-scale fusion correction is shown in Algorithm 1. Initially, the sonar image generated from the backscatter data is processed through steps such as the time-varied gain, median filtering, and pseudo-color transformation. Then, the sonar image is transformed into the HSV space. The NSST is decomposed in the feature *V* space that represents the brightness. MMSR theory is applied to estimate the illumination of the decomposed low-frequency sub-band image. In addition, the SDL method is adopted to suppress the noise of the high-frequency signal. The newly generated low- and high-frequency sub-bands are utilized to perform NSST inverse transformation and reconstruction to generate the enhanced brightness component *V'* space. Finally, the enhanced brightness *V'*, color component *H*, and saturation *S* are spatially inverse-transformed to generate an enhanced pseudo-color image.

---

**Algorithm 1** Multi-scale fusion strategy for side scan sonar image correction to improve low contrast and noise interference

---

**Input**: Read original image of side scan sonar. Set some constant item values, $\sigma_d$, $\sigma_r$, *wh*, $\Omega$, $\gamma$, *block*, *K*, *C*, *maxBlocks*, *iteration*. Select decomposer, decomposition direction, and level.

**Output**: Corrected sonar image

1: Pseudo-color processing of backscattered images;
2: Convert color image to HSV space and extract luminance component *V*;
3: Multi-scale decomposition of V component into low-frequency and high-frequency images is performed by NSST method;
4: Perform global mapping in low-frequency image to obtain image map $L_V$ by Equation (4);
5: Adopt bilateral filtering with sub-windows of different sizes to obtain initial illumination image LL by Equations (6)–(9);
6: Employ gamma correction strategy to obtain new illuminance image $LL_L$ by Equation (10);
7: Combine multi-scale retinex theory to obtain reflection image $L_R$ by Equation (3);
8: Multiply reflection image with new illumination image to obtain enhanced low-frequency image by Equation (11).
9: Estimate noise standard deviation of high-frequency images by Laplace filters;
10: Use DCT method to obtain initialization dictionary $D_0$;
11: Execute OMP sparse encoding and K-SVD dictionary process to iteratively update dictionary *D* and sparse matrix *X*;
12: Reconstruct denoised high-frequency images through new dictionaries and sparse matrices;
13: Multi-scale reconstruction of brightness feature image based on enhanced low-frequency image and denoised high-frequency images;
14: Inverted color space transformation into pseudo-color side scan sonar image;
15: Return enhanced sonar image and objective evaluation indexes.

---

## 3. Experiments and Results

### 3.1. Data Description and Parameter Settings

Klein3000 is a dual frequency side scan sonar developed by L-3 Company in the United States, representing a new technology in digital side scan sonar imaging. Klein3000 is employed to carry out underwater activities in the coastal area of the Pearl River Estuary. Different scene image sets are selected for the experimental analysis to verify the feasibility and effectiveness of the proposed correction strategy. In addition, the SonarWiz 7.12 software is applied to preprocess the acoustic backscatter signal and perform pseudo-color processing to form a sonar image with a spatial resolution of 0.2 m. Image pseudo-color processing is used to better identify target details and terrain contour features. Figure 6 illustrates that only four of these images are selected for the initial class of backscatter data, which fails to produce waterfall plots unless further adjustments are made. The measured dataset contains many typical scenarios, with prominent issues of low illumination and high noise, which meet the data requirements for sonar image correction.

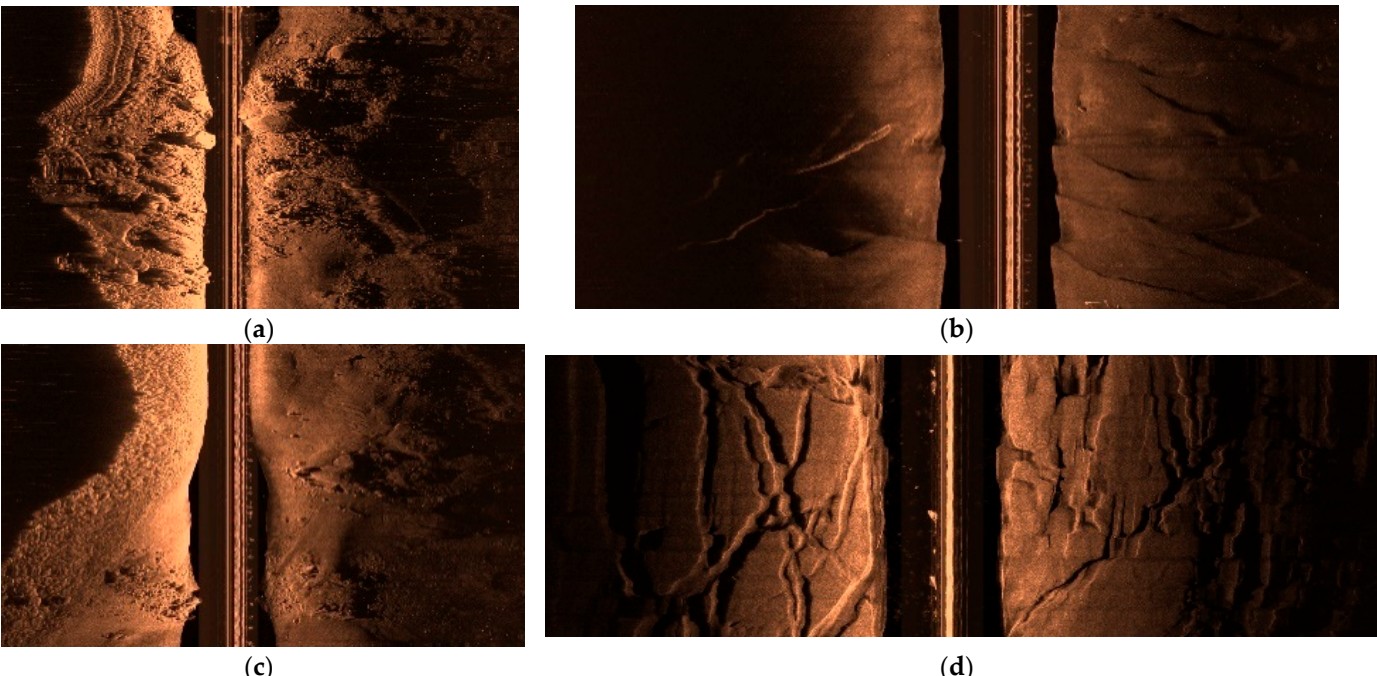

**Figure 6.** Measured sonar image set. (**a**) S1 image; (**b**) S2 image; (**c**) S3 image; (**d**) S4 image.

The suggested strategy for multi-scale and multi-directional image decomposition and reconstruction is based on the NSST. The scale decomposer is "maxflat". The orientation is a set of matrices [32, 32, 16], and the decomposition level is a set of matrices [2, 3, 4]. Furthermore, Table 1 displays the parameter selection of the low and high-frequency improvement techniques. The subsequent experimental analysis verifies the excellent performance of the parameters in Table 1.

**Table 1.** Parameter selection corresponding to improvement methods for low- and high-frequency sub-band images.

| Low Frequency | | High Frequency | |
|---|---|---|---|
| Distance STD | $\sigma_d = 3$ | Block size | $block = 8$ |
| Luminance STD | $\sigma_r = 0.1$ | Number of atoms | $K = 256$ |
| Kernel windows | $wh = 3$ | Limit factor | $C = 2.5$ |
| Neighborhood range | $\Omega = [5, 8, 15]$ | Maximum number of block samples | $maxBlocks = 260,000$ |
| Gamma factor | $\gamma = 1.8$ | Number of iterations | $iteration = 10$ |

*3.2. Objective Evaluation Metrics*

The correction of sonar images can be evaluated through subjective visual effects and objective indicators. Among them, subjective vision assesses the contrast primarily on the basis of features including brightness, the noise reduction effect, the clarity of the terrain texture, and local target shapes. Objective evaluation indicators are used to evaluate the quality of the correction and enhancement from various types of numerical statistical information. The objective indicators selected in the article include the average gradient (*AG*), standard deviation (*STD*), information entropy (*E*), and peak signal-to-noise ratio (*PSNR*) [37–39].

(1)　Average gradient

The average gradient reflects the brightness and blurriness of a side scan sonar image by representing the terrain texture changes and surrounding detail contrast elements. The larger the *AG* value, the stronger the sense of hierarchy and detail in the image. The formula is expressed as

$$
\begin{aligned}
AG &= \frac{\sum\limits_{x=1}^{w-1}\sum\limits_{y=1}^{h-1}\sqrt{\frac{F_x^2(x,y)+F_y^2(x,y)}{2}}}{(w-1)(h-1)} \\
F_x(x,y) &= I(x+1,y) - I(x,y) \\
F_y(x,y) &= I(x,y+1) - I(x,y)
\end{aligned}
\tag{12}
$$

where $I(x, y)$ is the grayscale value of point $(x, y)$. $F_x(x, y)$ and $F_y(x, y)$ represent the gradient changes in the horizontal and vertical directions of the sonar image.

(2)　Standard deviation

The standard deviation reflects the dispersion degree between the global grayscale and the average grayscale of the sonar image. The distribution of grayscale information is more extensive and the terrain texture is finer and richer with larger *STD* values. It is described as

$$
STD = \sqrt{\frac{\sum\limits_{i=1}^{w}\sum\limits_{j=1}^{h}(I(i,j)-\bar{I})^2}{w \times h}}
\tag{13}
$$

where $\bar{I}$ is the average grayscale value.

(3)　Information entropy

Information entropy, which can be applied to reflect the average amount of information in a sonar image, is the average amount of information contained in a particular grayscale value of an image. The amount of information in the corrected sonar image increases with a larger *E* value.

$$
E = -\sum_{i=0}^{255} num(i) \log num(i)
\tag{14}
$$

where *num* (*i*) is the ratio of the total number of pixels in the image to the number of statistics corresponding to pixel value *i*.

(4)　Peak signal-to-noise ratio

The purpose of the peak signal-to-noise ratio is to represent the effectiveness of denoising. The larger the value, the better the filtering effect and the clearer the sonar image. It can be described as

$$
PSNR = 10 \log_{10} \frac{I_{max}^2}{STD}
\tag{15}
$$

where $I_{max}$ is the maximum grayscale value of a sonar image.

### 3.3. Comparison of Background Color Effects

The S1 image in the measured dataset was selected to demonstrate the value of pseudo-color processing. Figure 7 illustrates how the varying backdrop colors of the images bring about distinct behaviors when using sonar correction approaches. The brightness level can be effectively improved by using the image correction technique shown in Figure 7a. Nevertheless, it is difficult to compare different correction methods for the backdrop and target in grayscale photographs with monochromatic backgrounds. Figure 7b offers enhanced approaches for contrast and correction, a stronger color intensity, and the better differentiation of human visual effects as compared to grayscale images. It is evident that the proposed strategy can improve the reef slope's scenery and successfully regulate the sand slope reef's brightness. By significantly darkening the target region, multi-area contrast control is achieved.

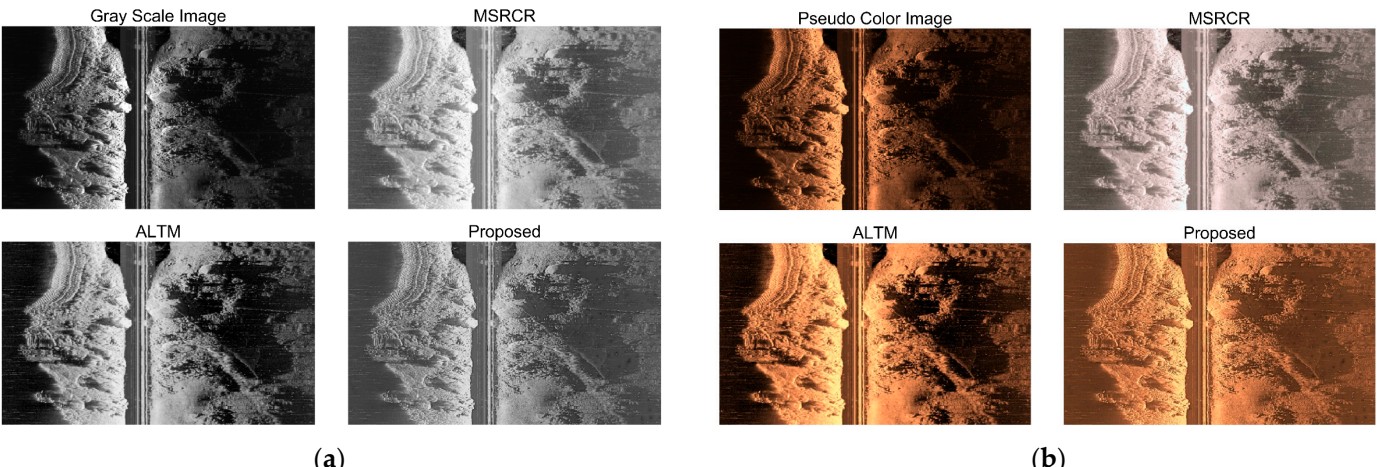

(**a**)        (**b**)

**Figure 7.** Correction effect on grayscale and pseudo-color sonar images. (**a**) Grayscale image effects; (**b**) pseudo-color image effects.

### 3.4. Optimal Parameter Selection for MMSR Model

The low-frequency signal decomposed by the NSST is a two-dimensional feature image. In order to demonstrate the optimal parameters of the MMSR model in Table 1, we conducted an experimental analysis on the window neighborhood range and gamma factor parameters.

#### 3.4.1. Range Selection of Window Neighborhood

We built single kernel and triple kernel windows for neighborhood estimation when estimating the illuminance component via bilateral filters. The neighborhood ranges $\Omega$ are matrix sets [5], [5, 8, 15], and [5, 10, 20], respectively.

Figure 8 displays the outcomes of the experiment. Figure 8a illustrates how the contrast can be somewhat enhanced by single core processing. As observed in Figure 8b, the image information becomes overexposed if the window neighborhood is too extensive. The overall improvement in the low-frequency images processed by the first two methods is insufficient. Figure 8c shows that the reasonable selection of neighborhood ranges for small, medium, and large windows can better balance the low-frequency signal information. The reliability of the neighborhood range selection in Table 1 is confirmed.

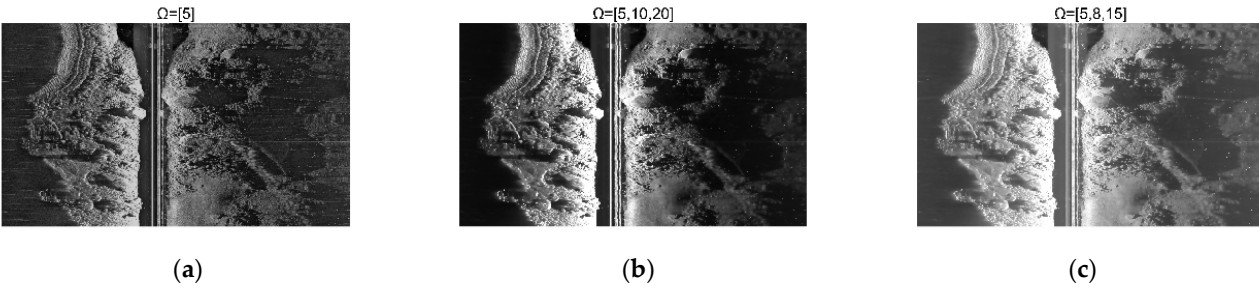

**Figure 8.** Low-frequency image enhancement effects in different neighborhood ranges. (**a**) Single kernel, Ω = [5]; (**b**) triple kernel, Ω = [5, 10, 20]; (**c**) triple kernel, Ω = [5, 8, 15].

### 3.4.2. Optimization of Gamma Factor

Gamma correction modifies the grayscale distribution of images using power-law functions, which can enhance the brightness of illuminated images to a certain amount. The contrast of high-grayscale areas needs to be improved because our side scan sonar image set is dark and contains speckle noise. The $\gamma$ must be configured to be larger than 1. We analyze the impact of the factor $\gamma$ on the enhancement of illumination images by fixing other preferred features in the table. Figure 9a shows the decomposed image at $\gamma = 1$, indicating the need for the further enhancement of the image information. The image in Figure 9c is overexposed, exhibiting gray and blurry features. The selected gamma factor in Table 1 has superior overall lifting performance, with prominent terrain and background information, as shown in Figure 9b.

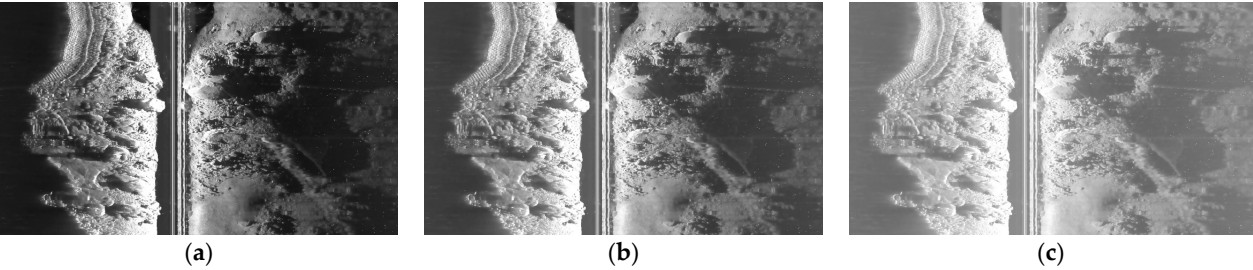

**Figure 9.** Low-frequency image enhancement effects under different gamma factors. (**a**) $\gamma = 1$; (**b**) $\gamma = 1.8$; (**c**) $\gamma = 3$.

### 3.5. Correction Effects of Various Retinex Models

Side scan sonar images with different targets as the main body reflect a great deal of information about underwater detection. In order to test the adaptability and universality of the proposed sonar image correction strategy, this section selects a set of measured side scan sonar images for experimental analysis, which includes sonar images of reefs, sand waves, gravel, and ship anchor marks as objects. The technical approaches of the retinex family, namely MSRCR, NPE, ALTM, and LIME, are contrasted with the suggested sonar correction procedure. The correction performance of each technology is evaluated via the subjective visual effect, as shown in Figure 10.

The side scan sonar image problems of low contrast and distributed speckle noise are depicted in the original image in Figure 10. Following MSRCR correction, the image is rather washed out and lacks the clear reflection of fine details. The target area exhibits overexposure and color deviation, as shown in the second line in Figure 10. Further work is required to enhance the NPE correction impact. The image is depicted in a grayscale manner overall, with the gray and hazy background areas typically including the scattered echo noise in the image.

The primary focus of the ALTM-processed waterfall image is easily discernible. The hue and contrast of the image are consistent with what is seen by humans. Target segmentation in the waterfall image becomes more challenging because of the image's overexposure

and the augmentation of the center region, which leads to an imprecise assessment of the mixing of sand waves and their undersides. The target edges, terrain texture, reef dispersion, and contrast are all improved in the LIME images. There is no way to reduce the horizontal wake echo signal. The noise echo dispersion in the middle of the waterfall image increases as it rises.

As a comparison, the suggested correction strategy emphasizes discrete detection information and well-defined contours for targets such as gullies, landscape textures, reefs, anchor marks, and target silhouette areas. Furthermore, the given solution is superior in terms of the overall image contrast, local prominent details, color distribution, and noise reduction. The proposed approach's universality is clearly illustrated by waterfall charts of targets with different sizes, brightness and grayscale levels, and terrain targets.

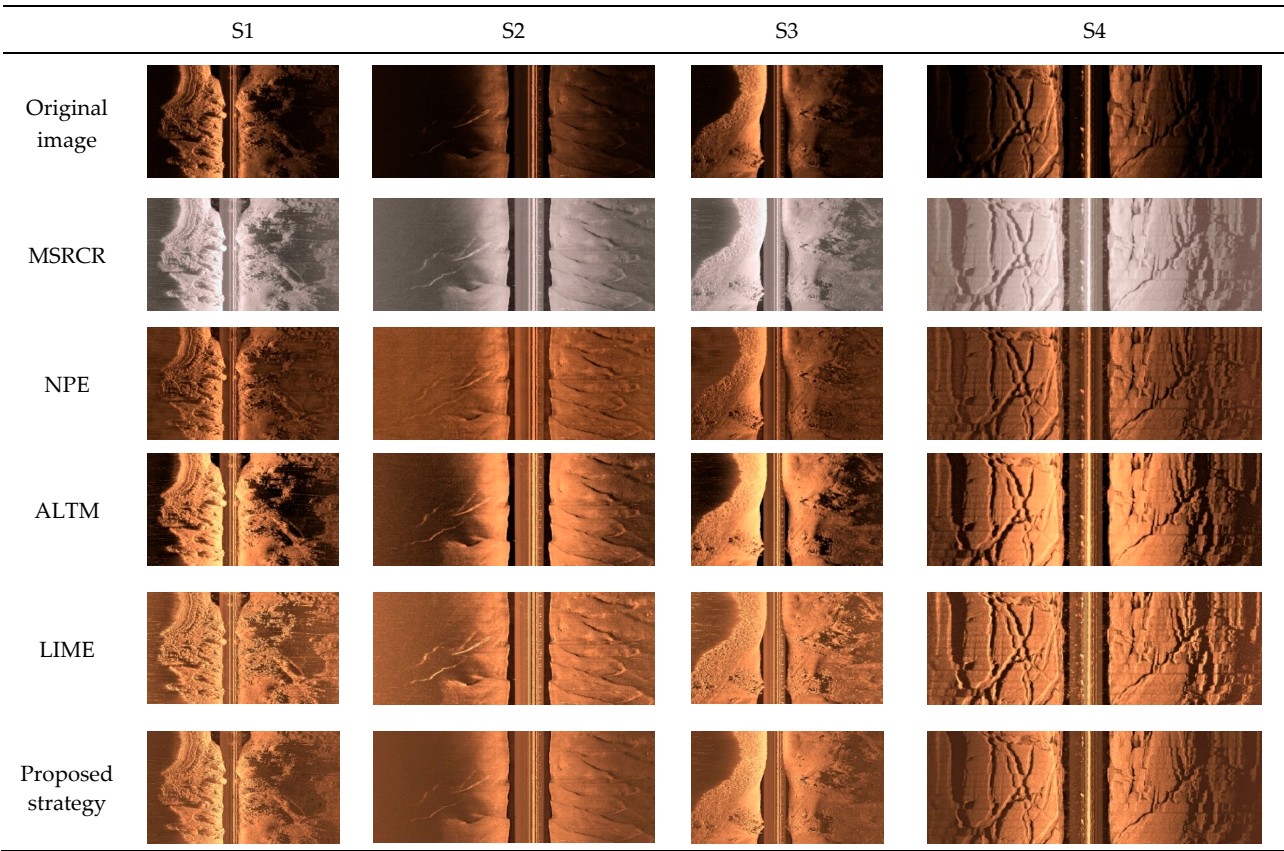

**Figure 10.** The correction effects of the various retinex models.

Further analysis was conducted on the correction performance of the proposed strategy and various retinex models at the levels of objective indicators, namely *STD*, *AG*, *E*, and *PSNR*, as shown in Table 2. Table 2 illustrates that none of the approaches, i.e., MSRCR, NPE, and ALTM, have attained ideal performance with any objective indicator, and the values of many indicators fluctuate substantially. The LIME technique is used to process the image with the highest *AG* value. Nevertheless, the *AG* value of the suggested strategy also falls within the upstream level and somewhat represents the image clarity. Furthermore, the *PSNR* value indicates that the suppression effect of LIME on scattered noise signals is significantly diminished, which contradicts the idea that low illumination and noise issues need to be improved simultaneously. The waterfall images S1–S4 processed with the suggested technique exhibit a rise in the *STD* values of at least 13 points, which accurately represent the rich and delicate geographical texture as well as the larger dispersion of the corrected grayscale information. In addition, the *PSNR* index value reflects the effective weakening of scattering spots and wake noise by our method, which is far superior regarding the performance of the other methods.

**Table 2.** Objective evaluation indicators for the retinex variant models.

| Image | Metric | MSRCR | NPE | ALTM | LIME | Processed Strategy |
|---|---|---|---|---|---|---|
| S1 | *STD* | 53.45 | 41.43 | 51.03 | 64.04 | 69.35 |
| | *AG* | 9.94 | 10.52 | 13.19 | 17.15 | 14.14 |
| | *E* | 5.62 | 6.89 | 7.20 | 7.45 | 7.46 |
| | *PSNR* | 7.57 | 12.06 | 11.27 | 8.93 | 18.30 |
| S2 | *STD* | 40.93 | 27.16 | 33.44 | 51.58 | 63.34 |
| | *AG* | 6.48 | 8.94 | 8.72 | 11.19 | 10.88 |
| | *E* | 4.62 | 6.64 | 6.67 | 7.05 | 7.10 |
| | *PSNR* | 7.51 | 10.54 | 10.11 | 8.07 | 11.14 |
| S3 | *STD* | 49.99 | 47.55 | 38.39 | 62.88 | 65.90 |
| | *AG* | 8.18 | 9.16 | 11.15 | 14.81 | 12.17 |
| | *E* | 5.75 | 6.84 | 7.29 | 7.41 | 7.51 |
| | *PSNR* | 7.24 | 10.84 | 11.42 | 8.78 | 17.78 |
| S4 | *STD* | 45.33 | 34.67 | 51.58 | 63.61 | 63.61 |
| | *AG* | 6.33 | 8.12 | 8.84 | 13.39 | 9.09 |
| | *E* | 4.99 | 6.95 | 7.31 | 7.30 | 7.53 |
| | *PSNR* | 5.32 | 9.62 | 11.25 | 8.52 | 14.51 |

The grayscale statistical regularity is a quantitative method of analyzing corrected images' quality. Figure 11 shows the grayscale distribution characteristics of each waterfall plot. Every sonar image possesses a discontinuous distribution of low grayscale values. MSRCR, BPDHE, and ALTM are unable to guarantee constant changes in grayscale information and instead exhibit sporadic step distributions, especially with BPDHE having a large fault span. Although NPE shows an almost perfect balance, the great visual stretching effect of the contrast is limited by the disproportionate enlargement of its maximum grayscale value. While LIME guarantees a balanced concentration and constant change in grayscale values, there is a problematic phenomenon of abrupt stretching at the maximum grayscale value. The image distribution as corrected by the proposed strategy indicates that the measured image set conforms to the $\chi^2$ characteristics. Extreme abrupt signals, such exposure and dimming, are absent from the corrected grayscale values at both ends. The entire grayscale is steadily improved and adjusted to the right.

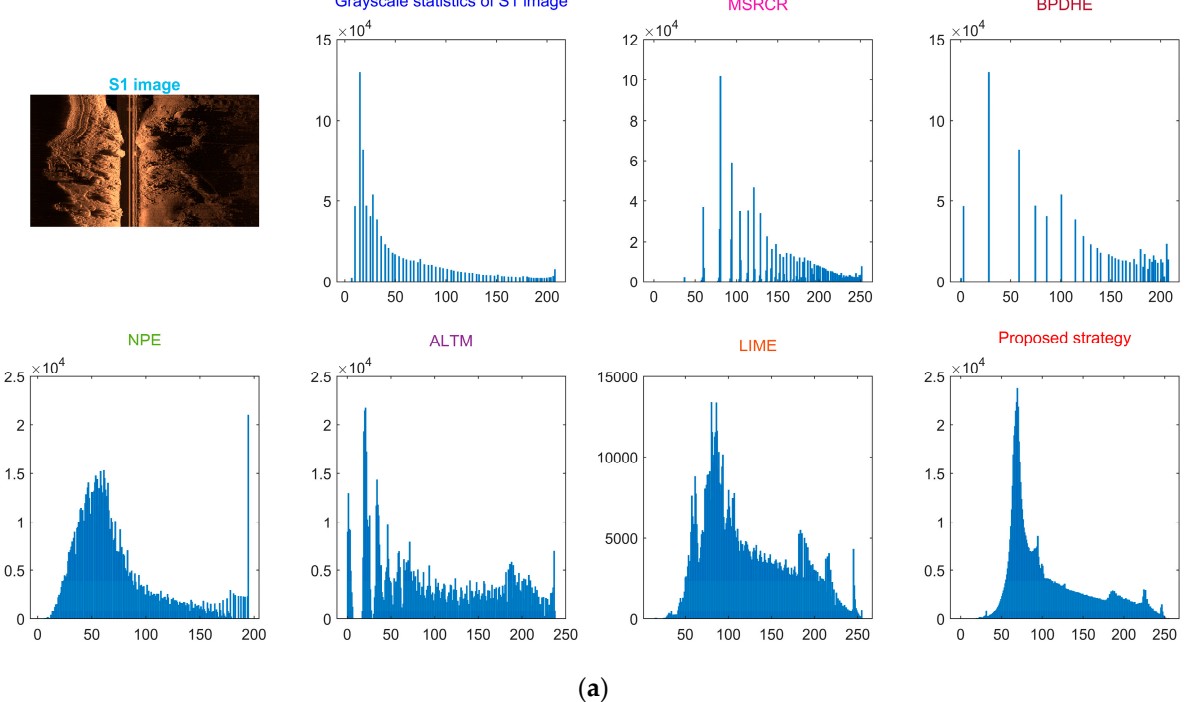

(a)

**Figure 11.** *Cont.*

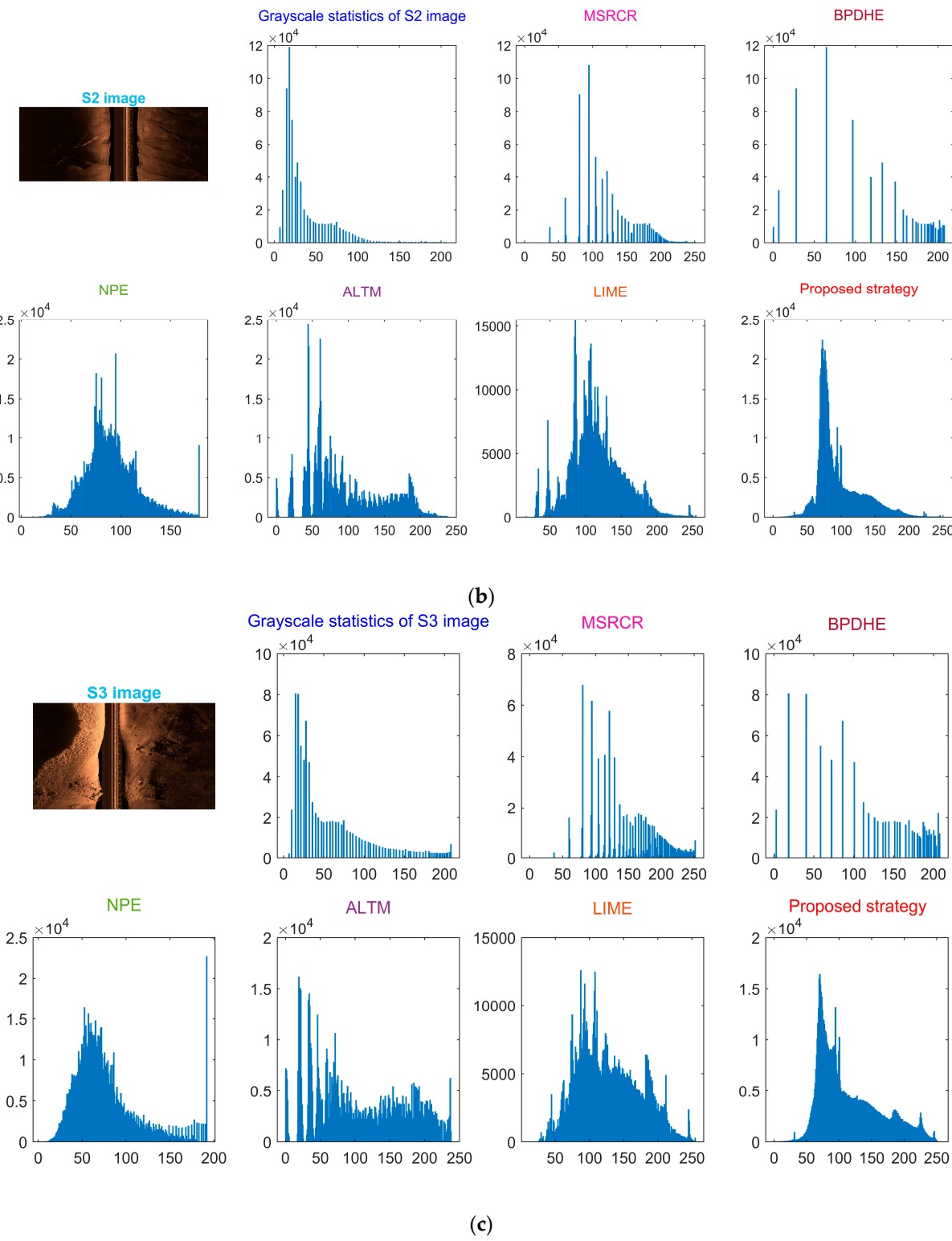

**Figure 11.** *Cont.*

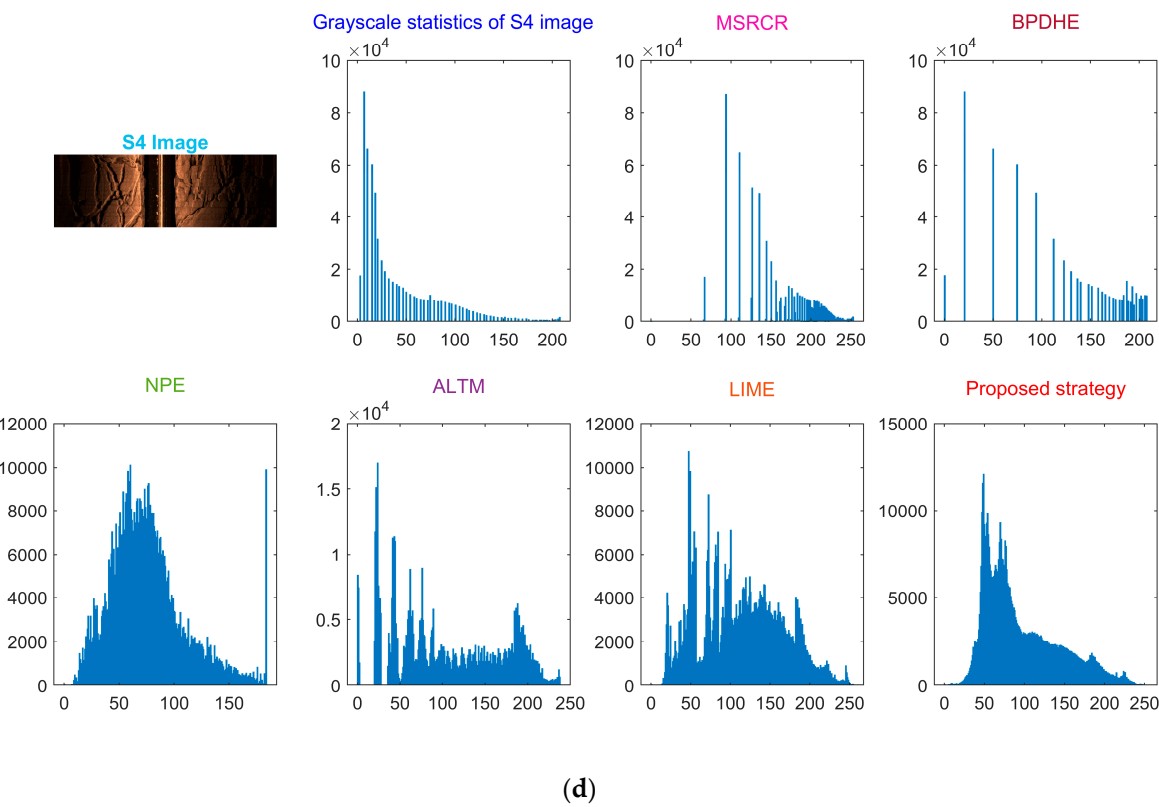

(**d**)

**Figure 11.** The grayscale distribution characteristics of the retinex variant models. (**a**) S1 image; (**b**) S2 image; (**c**) S3 image; (**d**) S4 image.

### 3.6. Comparative Effects of Other Correction Techniques

Additionally, we selected four representative side scan sonar calibration methods, BPDHE, FE, WT, and TVRLRA, for comparison using the measured dataset, as depicted in Figure 12. After correction with the BPDHE and WT techniques, Figure 12 demonstrates that the bright area in the middle of the image is overexposed, clearly displaying the white halo phenomenon. The image that was produced following FE processing exhibits a low overall grayscale elevation. The terrain's backdrop sections and sand slopes are locally darker following TVRLRA correction. The suggested method performs image correction with strong overall contrast, modest color alterations, and distinct local features and borders.

Subjective visual observation of the corrected image may not be sufficient to distinguish these methods. Table 3 displays the four objective metrics used to analyze the technical performance. The *STD* values of the FE approach are comparatively low in Table 3, indicating a concentrated and dark grayscale distribution of information. The *E* and *PSNR* values of BPDHE are below one order of magnitude, revealing the problem of low average information content and high noise levels. The comparatively low *AG* value of TVRLRA reflects the lack of effective detection and enhancement of some micro-change information. Furthermore, the *PSNR* index demonstrates that the suggested strategy may achieve noise reduction on par with the advanced TVRLRA. The suggested approach can successfully reduce the spots and salt-and-pepper noise in the sonar image of the submodule, while simultaneously enhancing the contrast and clarity of the entire image.

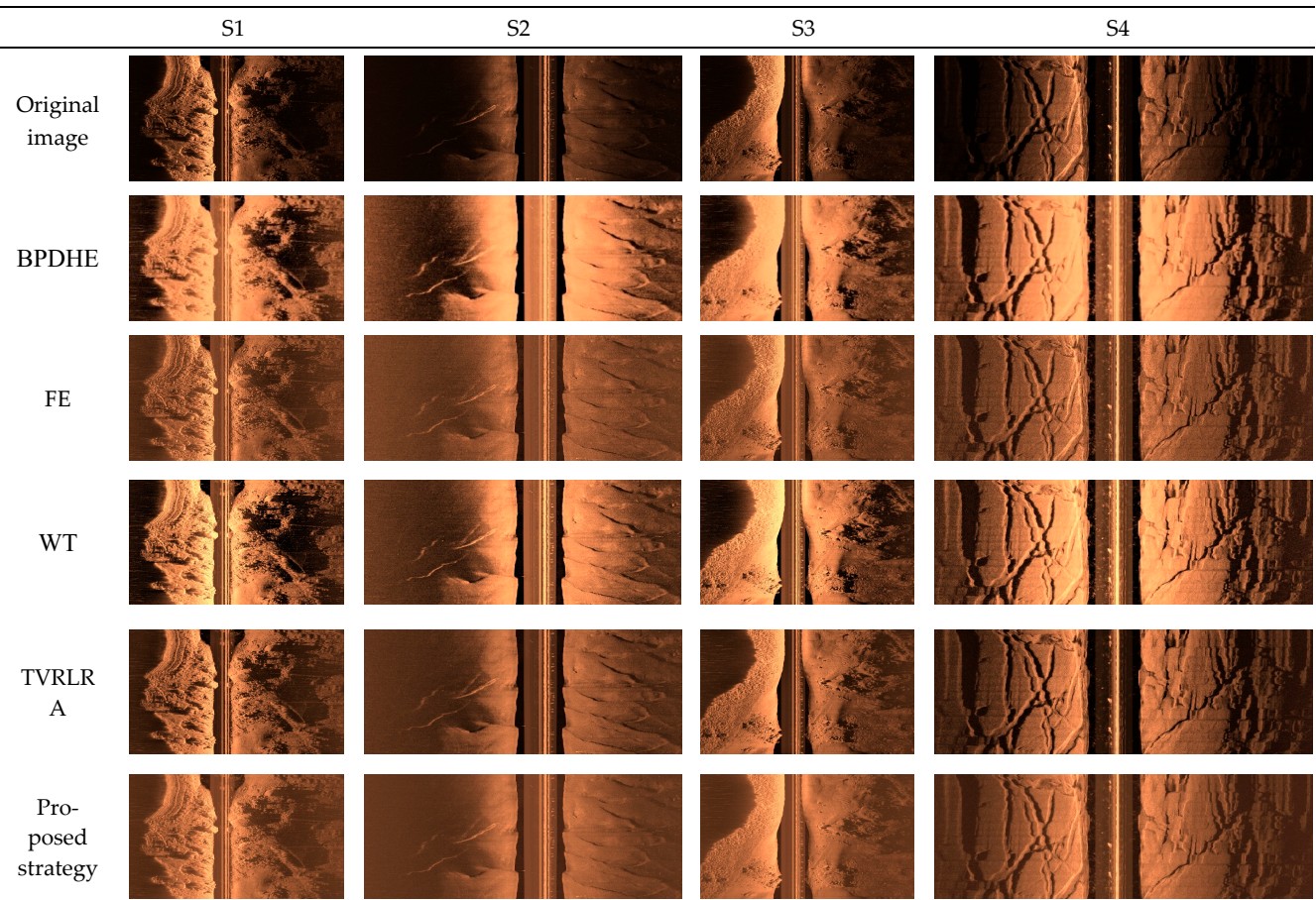

**Figure 12.** The comparative effects of representative correction techniques.

**Table 3.** Objective evaluation indicators for the representative correction techniques.

| Image | Metric | BPDHE | FE | WT | TVRLRA | Processed Strategy |
|-------|--------|-------|-----|-----|--------|--------------------|
| S1 | *STD* | 50.87 | 39.23 | 65.28 | 45.50 | 69.35 |
|    | *AG* | 11.67 | 11.17 | 12.31 | 9.36 | 14.14 |
|    | *E* | 4.63 | 7.02 | 7.46 | 7.10 | 7.46 |
|    | *PSNR* | 9.89 | 15.11 | 12.06 | 16.81 | 18.30 |
| S2 | *STD* | 36.41 | 24.67 | 48.80 | 29.73 | 63.34 |
|    | *AG* | 6.63 | 7.31 | 8.21 | 5.24 | 10.88 |
|    | *E* | 4.10 | 6.47 | 7.10 | 6.50 | 7.10 |
|    | *PSNR* | 7.50 | 10.51 | 10.11 | 10.58 | 11.14 |
| S3 | *STD* | 45.01 | 36.37 | 62.08 | 43.35 | 65.90 |
|    | *AG* | 10.06 | 9.62 | 10.42 | 7.94 | 12.17 |
|    | *E* | 4.79 | 7.04 | 7.51 | 7.20 | 7.51 |
|    | *PSNR* | 10.28 | 14.98 | 11.42 | 16.13 | 17.78 |
| S4 | *STD* | 47.20 | 35.76 | 59.92 | 43.29 | 63.61 |
|    | *AG* | 8.59 | 8.42 | 8.50 | 6.41 | 9.09 |
|    | *E* | 4.50 | 6.96 | 7.31 | 7.06 | 7.53 |
|    | *PSNR* | 8.35 | 14.38 | 9.62 | 14.46 | 14.51 |

## 4. Discussion

### 4.1. Comparison of Filter Performance

We employ guided filtering to investigate the impact of the filter selection on the technology, taking into account the utilization of bilateral filtering in the suggested approach. As displayed in Figure 13, the filtering impact is assessed using the PSNR index. The visual

contrast is limited to a certain degree and exhibits partial scattered speckle noise in Figure 13b. Bilateral filtering highlights textural information while safeguarding the target margins, as seen in Figure 13c. Furthermore, a larger PSNR value reveals that bilateral filtering weakens the influence of noise on image correction to a certain extent. The corrected wake speckle noise is suppressed and the overall contrast saturation of the image is enhanced.

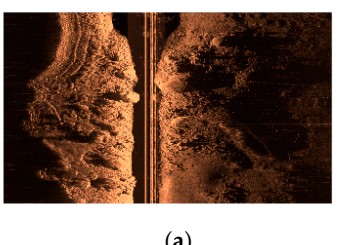 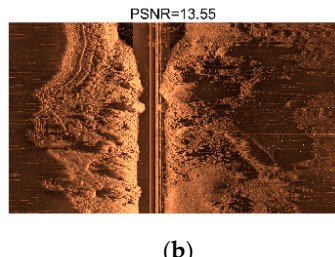 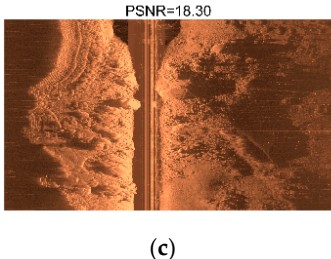

(**a**)  (**b**)  (**c**)

**Figure 13.** Correction effects of two filters. (**a**) Original; (**b**) guided filtering; (**c**) ours—bilateral filtering.

### 4.2. Expansion of Our strategy

To verify the excellent performance of the proposed strategy, we selected three publicly available datasets and one sonar image after radiometric correction for experimentation. The sonar image correction results obtained from different scenes, sources, and measurement conditions are shown in Figure 14. Figure 14 shows the correction effects of the six comparison techniques.

Figure 14 reveals the limited capabilities of BPDHE correction. LIME excessively enhances the contrast in both the non-pseudo-color sonar image and forward-looking sonar image. However, LIME encounters correction bias in the other two images. This phenomenon indicates that the adaptability of the LIME technique in side scan sonar image correction still needs to be discussed. The overall darkening of the image after advanced TVRLRA correction is supported by the literature [8]. The non-color images processed by WT and FE exhibit a degree of hierarchical blur and visible speckle noise. Overall, our proposed strategy has significant advantages in denoising various types of sonar images, highlighting detailed features and ensuring the image contrast and clarity to a certain extent.

The objective evaluation values to promote sonar image collection are displayed in Table 4. The suggested approach performs better than the state-of-the-art FE and VRLRA approaches, according to the *PSNR*. The *AG*, *STD*, and *E* values reflect that the proposed strategy can ensure the information content, continuous grayscale features, and clear hierarchy of the image during the correction process. In addition, the reflected texture, detail changes, and other aspects are superior.

**Table 4.** Objective evaluation indicators for promotion of sonar image sets.

| Image | Metric | BPDHE | FE | WT | TVRLRA | ALTM | LIME | Processed Strategy |
|---|---|---|---|---|---|---|---|---|
| P1 | *STD* | 25.06 | 30.42 | 32.32 | 25.66 | 32.32 | 37.38 | 41.70 |
| | *AG* | 11.29 | 13.91 | 13.06 | 9.94 | 13.06 | 18.83 | 18.11 |
| | *E* | 6.34 | 6.85 | 6.93 | 6.59 | 6.93 | 6.83 | 7.24 |
| | *PSNR* | 12.23 | 15.53 | 13.98 | 14.97 | 13.98 | 7.45 | 16.48 |
| P2 | *STD* | 42.99 | 60.5 | 59.87 | 51.16 | 59.87 | 54.4 | 62.17 |
| | *AG* | 7.37 | 11.03 | 11.29 | 7.31 | 11.29 | 14.73 | 15.99 |
| | *E* | 4.56 | 5.7 | 5.74 | 5.59 | 5.74 | 5.7 | 5.93 |
| | *PSNR* | 9.41 | 12.57 | 12.32 | 16.71 | 12.32 | 8.49 | 17.87 |
| P3 | *STD* | 49.53 | 49.04 | 60.65 | 53.71 | 60.65 | 63.07 | 65.78 |
| | *AG* | 10.62 | 12.28 | 10.71 | 10.94 | 10.71 | 10.84 | 11.89 |
| | *E* | 7.24 | 7.05 | 7.21 | 7.31 | 7.21 | 7.33 | 7.97 |
| | *PSNR* | 10.58 | 17.96 | 10.5 | 17.38 | 10.5 | 11 | 18.07 |
| P4 | *STD* | 62.79 | 64.86 | 72.17 | 67.53 | 73.33 | 77.18 | 68.01 |
| | *AG* | 22.20 | 24.53 | 23.55 | 22.49 | 26.17 | 37.81 | 39.90 |
| | *E* | 6.54 | 7.54 | 7.59 | 7.5 | 7.59 | 7.71 | 7.88 |
| | *PSNR* | 18.21 | 22.25 | 13.1 | 24.76 | 21.38 | 12.02 | 27.51 |

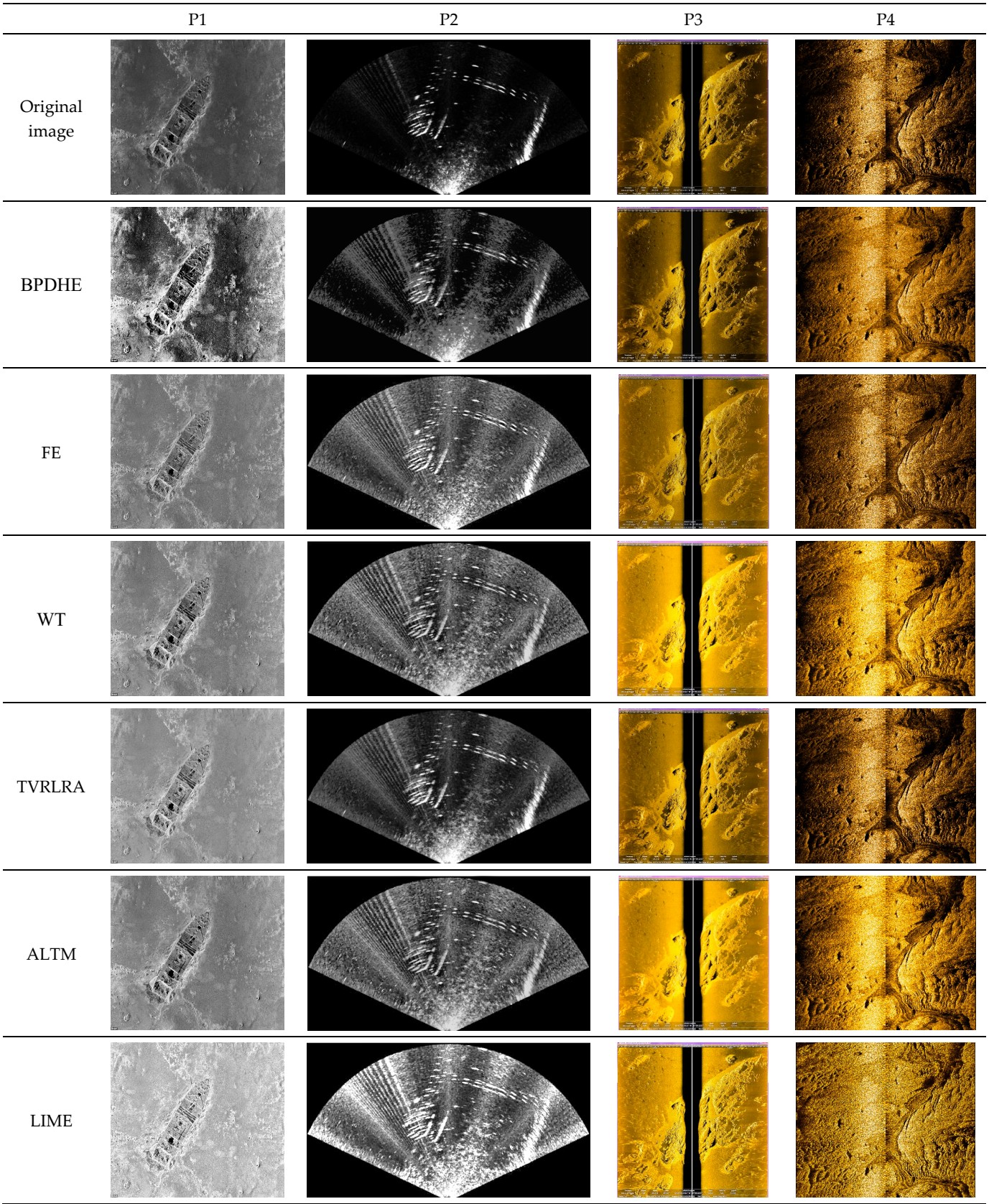

**Figure 14.** *Cont.*

| | | | |
|---|---|---|---|
| Proposed strategy | 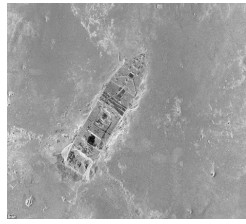 | 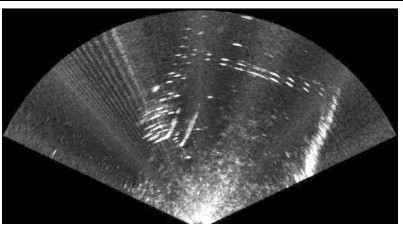 | 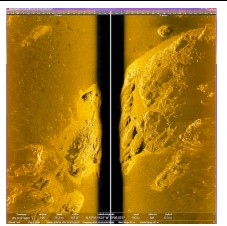 | 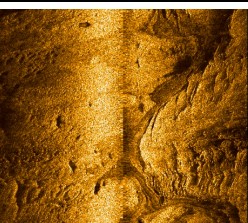 |

**Figure 14.** The effectiveness of different correction methods for multi-source scenes.

## 5. Conclusions

This paper proposes a multi-scale fusion strategy to address the issues of noise filtering and low contrast enhancement in side scan sonar images. The excellent quality, effectiveness, and universality of the proposed strategy are validated using measured sonar images and publicly available datasets. The experiment comprehensively considers pseudo-color processing, the window neighborhood range, the gamma factor, filter selection, and correction techniques for different classifications and analyses. The experimental results indicate that pseudo-color methods can more accurately represent the corrective effect to a certain extent. Low-frequency signal enhancement and noise reduction can be successfully guaranteed by the ideal parameters of the MMSR model. Compared with the retinex variant model and existing advanced sonar image correction methods, the proposed strategy successfully achieves advantages such as overall image contrast enhancement, clear contours, full color, and a clear hierarchy. The findings of the grayscale histogram demonstrate that the suggested approach may successfully guarantee ongoing modifications following adjustment and grayscale improvement, which is in line with the distributional properties of the sonar image. In addition, objective indicators reveal that the proposed strategy exhibits significant noise filtering characteristics and excellent fidelity compared to eight correction methods: BPDHE, MSRCR, NPE, ALTM, LIME, FE, WT, and TVRLRA. The suggested method improves the STD, E, and PSNR in a variety of scene images by at least 8%, 4.5%, and 6%, respectively. The universality is thoroughly proven by the promotion tests conducted on public datasets. As expected, we find that the corrected side scan sonar image contains local speckle noise. This may be attributed to our HSV spatial transformation, which only processes the brightness feature V component. Our next study will aim to thoroughly investigate the noise characteristics of numerous channels in the future.

**Author Contributions:** Methodology, P.Z. and J.C.; Formal analysis, J.C.; Funding acquisition, J.G. and H.Z.; Data curation, P.T.; Writing—original draft preparation, P.Z. and J.C. All authors have read and agreed to the published version of the manuscript.

**Funding:** This study was supported by the National Natural Science Foundation of China (Grant: 42162025) and the Key Science and Technology Project of the Jiangxi Provincial Department of Water Resources (Grant: 202124ZDKT21).

**Data Availability Statement:** The data supporting this study are available from the authors. Some publicly available sonar datasets were sourced from Advanced Studio: Digital Fabric for the Arts, literature references (A Dataset with Multibeam Forward Looking Sonar for Underwater), and side scan sonar example images from the EdgeTech official website.

**Acknowledgments:** Thank you very much to Achan from Chesapeake Technology for providing the image preprocessing software, which made important contributions to the preliminary experiment.

**Conflicts of Interest:** The authors declare no conflicts of interest.

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
