# Peer review of "A Multi-Scale Fusion Strategy for Side Scan Sonar Image Correction to Improve Low Contrast and Noise Interference"

_remotesensing, doi:10.3390/rs16101752_

Round 1

Reviewer 1 Report

Comments and Suggestions for Authors

 This paper presents a method for noise processing and low-contrast enhancement of sonar images based on HSV color space decomposition and reconvergence. The authors describe the details of the method implementation and conduct a more comprehensive validation and comparison analysis. Some comments are list as follows.

Comment 1: A comprehensive, categorized review of side-scan sonar image correction is presented in the paper. It is recommended that the authors add literature citations for the specific methods listed in each category in Figure 1 for readers' convenience.

Comment 2: What are the differences in image processing between optically-based images and acoustically-based sonar side-scan images? What is the difficulty in processing sonar side scan images? Please give more details.

Comment 3: Does a higher value for an objective evaluation metric mean better performance? It is recommended that the authors state this in the text.

Comment 4: The NSST decomposition method and SDL denoising method involved in the introduction of the methods in the second part of the paper are relatively mature methods; a brief overview by the authors is sufficient. The proposed modified MMSR model can be introduced in detail to highlight the main innovative work of the authors.

Comment 5: The method proposed in the paper involves the selection of some parameters, as shown in Table 2 of the paper. Do all these parameters affect the quality of the final image, and which are the ones that have a greater impact on the results? It is suggested that appropriate explanations be given in the paper.

Comment 6: Similar to question 5, does the proposed method rely too much on professional experience to make the appropriate parameter selection to achieve better sonar image enhancement? Do the authors consider that the method introduces too many parameter settings while achieving image enhancement?

Comment 7: Both the luminance enhancement and the scattering noise treatment in the proposed method are carried out in the V channel. Although the brightness enhancement depends on the V channel, the noise may be related to other channels, which may affect the noise reduction effect. It is suggested that the authors make further comparisons.

Comment 8: The conclusion should be improved. I would like to suggest a summary with a quantitative description of your proposed method as much as possible.

Comment 9: More in-field applications should be considered to testify the proposed method. Current contents cannot show evidence of the proposed method' strength on different bridges.

Author Response

Please find the attached reply for detailed modification comments, thank you.

Reviewer 2 Report

Comments and Suggestions for Authors

(1) In the abstract, the author stated that the proposed method is suitable for side-scan sonar images under different conditions, but the different conditions were not reflected in the subsequent experiments and analysis of experimental results. It is recommended to add some experimental results data at the end of the abstract, such as "It is xx% ahead of the current state-of-the-art method xxx"

(2) In the introduction, add some discussion about the cited papers instead of simply listing the methods and contributions of other researchers.

(3) In Chapter 2, the specific details of the implementation of the overall method should be added. The author only introduces the content of the three proposed parts, but the overall method should be summarized again. At the same time, the author said in 2.1 that the paper combines multi-scale fusion and retinex theory. However, I think the method used in the article combines multi-scale parts is somewhat missing. I hope the author can explain this part in detail.

(4) This article has done relevant experiments, but it is recommended to add some open source data sets and use pictures from the open source data sets for testing to better prove the superiority of the method and make the results more convincing.

(5) The content of 2.5 should be added to Chapter 3, which will help readers understand the content of the article.

(6) An explanation of Figure 11 should be added, the data in the figure needs to be explained, and the reasons for doing this part of the experiment should be explained.

(7) Most of the contrast methods in contrast experiments are low-illumination image methods, and there are almost no methods applied in side scan sonar. It is hoped that some related contrast experiments in side scan sonar can be added to improve the results. More convincing. At the same time, references of relevant methods are added to the comparative experiments to facilitate readers to quickly understand the articles on comparative methods.

(8) In order to ensure that readers can reproduce the work of this article, it is recommended to add some experimental details, such as algorithm steps or pseudocode, to Chapter 3.

(9) The author stated in his contribution that the proposed method can weaken the impact of noise interference, but this contribution was not reflected in the experiment.

(10) It is recommended that the author add a summary of the overall experimental content and results in the experimental discussion section, and analyze some problems that occurred or may arise during the experiment.

(11) The format of the paper needs to be modified. For example, the full name of SSS in line 50 is not given; there is a problem with the format of Figure 9; the title and table of Table 3 are not on the same page, etc. There are many similar problems that need to be revised carefully. The format of Figures 11 and 13 needs to be reformatted.

Author Response

(The authors gave the same response as above.)

Round 2

Reviewer 1 Report

Comments and Suggestions for Authors

The authors have made comprehensive responses to the questions raised last time as well as revisions, and the quality of the paper has improved considerably. There are still a few modifications to be made:

Comment 1: It is recommended that the authors further condense the significance of the study to highlight the contribution made by the article.

As the author states this paper proposes a strategy of integrating multiple technologies to improve the low illumination and high noise issues in side scan sonar images.

But I am curious about what is the further purpose? How this research contribute to the further potential engineering practices? If only for underwater target detection, for my point of view, current methods are enough for target identification, i.e. to identify what the target is. In other words, only target detection may not require high resolutions images. So it is recommended that the authors highlight their research’s unique/necessary contribution from the point of engineering needs.

Comment 2: Results and discussion on Table 3 should be improved. From Table 3, I cannot see a clear improvement of the authors’ proposed method. For instance, AG value shows LIME method seems better, is it?

Comment 3: Can the author give a clearer answer to “What are the differences in image processing between optically-based images and acoustically-based sonar side-scan images? What is the difficulty in processing sonar side scan images?”

Comment 4: The concluding section suggests a multi-point summary and, where possible, quantitative enhancement indicators to highlight the strengths of the methodology.

Comment 5: Authors are advised to check the full text after the framing of the article has been adjusted, e.g. lines 153-154.

Comment 6: It is suggested that the author submit a trackable revised version with only modified contents be highlighted, not including format revision recordes.

Author Response

We have made the modifications according to your feedback. Your proposal is very helpful for us to polish our paper again. Please refer to the attachment for detailed response comments., Thank you.

Reviewer 2 Report

Comments and Suggestions for Authors

This article proposes NSST, MMSR, SDL and other methods, and describes each processing step in detail. In response to the questions raised, the author revised the experimental part, abstract, introduction, etc. in the manuscript, added experimental details, and enriched the experimental content to facilitate readers' learning and reproduction.

Author Response

Thank you very much for your review. Your feedback provides guidance for the viewpoints and writing of our paper. By the way, we have revised the relevant abstract, introduction, partial experimental description, and quantifiable conclusion description. Thank you very much for your guidance recently.
